# Psychometric properties of End Stage Renal Disease-Adherence Questionnaire-Sinhalese version among patients receiving haemodialysis

Chalani Lasanthika[1,2]*, Kamani Wanigasuriya[3], Usha Hettiaratchi[4], Thamara Dilhani Amarasekara[2], Christine Sampatha Evangeline Goonewardena[5]

1 Faculty of Graduate Studies, University of Sri Jayewardenepura, Nugegoda, Sri Lanka, 2 Department of Nursing and Midwifery, Faculty of Allied Health Sciences, University of Sri Jayewardenepura, Nugegoda, Sri Lanka, 3 Center for Kidney Research, Faculty of Medical Sciences, University of Sri Jayewardenepura, Nugegoda, Sri Lanka, 4 Department of Biochemistry, Faculty of Medical Sciences, University of Sri Jayewardenepura, Nugegoda, Sri Lanka, 5 Department of Community Medicine, Faculty of Medical Sciences, University of Sri Jayewardenepura, Nugegoda, Sri Lanka

* c.nimna@sjp.ac.lk

**Data Availability Statement:** All relevant data are within the paper and its Supporting information files.

## Abstract

The chronic kidney disease is a leading public health concern, particularly in low-to middle-income countries, while the number of patients receiving haemodialysis is rapidly increasing. Adherence to a complex treatment regimen is vital for those patients on maintenance haemodialysis though the precise evaluation is reported inadequately. This study aimed to evaluate the psychometric properties of Sinhalese version of End Stage Renal Disease-Adherence Questionnaire (SINESRD-AQ) to assess treatment adherence behaviour among patients receiving haemodialysis in a Sri Lankan hospital. The cultural adaptation of ESRD-AQ involved forward and back translation, expert committee consolidation and pretesting among patients (n = 10). Face and content validity of the questionnaire was evaluated using a modified Delphi technique. Construct validity of the subscales of SINESRD-AQ was evaluated using confirmatory factor analysis (CFA). A descriptive cross-sectional study among a consecutive sample of 150 patients receiving haemodialysis in a selected Teaching hospital, Sri Lanka was involved in performing CFA. Reliability was confirmed with test-retest reliability. Excellent face and content validity were reported with Item level content validity index (0.83–1.0), Average Item level content validity index for whole scale (0.93), Item level content validity ratio (0.67–1.0) and modified kappa statistic coefficient (0.81–1.0). CFA of two subscales demonstrated better indices closure to the model fit with five-item two factor model for direct adherence behaviour subscale and eight-item two factor model for attitude/perception subscale. The intra class correlation coefficient of 0.837 (p<0.001) and 0.752 (p<0.001) indicated acceptable test-retest reliability of direct adherence behaviour and attitude/perception subscale respectively. The study confirmed SINESRD-AQ as a valid and reliable measure which enables periodic assessment of treatment adherence behaviour of the patients receiving haemodialysis in a Sri Lankan hospital.

**Funding:** The study was funded by the University Research Grant of University of Sri Jayewardenepura, Sri Lanka (Grant No-ASP/ 01/ RE/MED/2017/61) received by CSEG. The funder had no role in study design, data collection and analysis, decision to publish, or preparation of the manuscript. URL of funder's website - www.sjp.ac. lk.

**Competing interests:** The authors have declared that no competing interests exist.

## Introduction

Chronic kidney disease (CKD) is an emerging public health problem with a high global burden of disease [1]. End stage renal disease (ESRD) is the terminal stage of CKD with severely declined glomerular filtration (GFR<15ml/min) [2] which require renal replacement therapy either long term dialysis or renal transplantation for survival of the patients [3].

Nonetheless, haemodialysis has become as the most demanding and practical treatment modality among patients with ESRD in Sri Lanka due to resource constraints and shortage of kidney donors for transplantation [4]. The number of haemodialysis sessions carried out in National Hospital of Sri Lanka was dramatically increased from approximately 4600 in 2003 to more than 11,600 in 2013 with an average of thousand haemodialysis sessions per month [5].

Adherence to the prescribed dialysis treatment regimen including regular haemodialysis attendance, prescribed medications, recommended diet and fluid restrictions is vital for better outcomes of the patients on maintenance haemodialysis [6]. The patient's ability to comply with prescribed treatment regimen depends on the understanding and motivation of the patients. Nevertheless, approximately 50% of patients with ESRD receiving haemodialysis showed non-adherence to their prescribed treatment regimen [7].

Significant variations of reported non-adherence rates have been observed due to lack of reliable measurement tools that address all the facets of treatment adherence behaviour of the patients receiving haemodialysis. Of the variety of measures, clinical measures such as biological markers (interdialytic weight gain) and biochemical markers (serum creatinine, blood urea nitrogen, serum potassium and phosphate levels etc.) can be viewed as objective measures while a valid and reliable tool to assess subjective phenomenon was in dearth in literature [8]. In particular, a few multidimensional, self-reported adherence scales have been developed and tested to assess treatment adherence behaviours among patients receiving haemodialysis worldwide.

End Stage Renal Disease Adherence Questionnaire (ESRD-AQ) is one of the widely accepted multidimensional instruments designed by Kim et al. in Los Angeles, California to assess self-reported treatment adherence behaviours in patients receiving haemodialysis [8]. It consists of two subscales namely 'Direct adherence behaviour subscale' and 'Attitude/perception subscale' which are directly contributed in measuring adherence behaviours. ESRD-AQ has been culturally adapted and validated into different languages and used to assess treatment adherence among patients receiving haemodialysis in Portugal, United states, Spain, Brazil, Egypt and Palestine [6, 8–12].

With the increasing prevalence of patients with CKD receiving haemodialysis in Sri Lanka, a valid and reliable, multidimensional instrument is required to determine the degree of adherence to the haemodialysis treatment regimen among Sinhala speaking patients. The assessment of treatment adherence enables addressing any deficiencies identified to prevent the occurrence of adverse events of the disease among patients. As the increasing prevalence of ESRD has been placed significant economic burden on the Sri Lankan health care sector [4], patient adherence to their own treatment might help to reduce the health care cost and also their own expenditure for the treatments. However, there is limited evidence available on self-reported adherence among patients receiving haemodialysis in Sri Lankan hospitals.

Among the previous studies conducted in Sri Lanka, none have used ESRD-AQ to assess self-reported adherence and no other Sinhalese version of instruments measuring adherence is available for use among Sinhala speaking patients receiving haemodialysis. Thus, availability of an instrument from Sinhala language is immensely worth. Besides, making periodic assessments on self-reported adherence will be an extra motivation for the patients to pay their strong attention on following medical advice as well as to make adjustments of the care

provided by the health caregivers. Thus, the aim of the study was to evaluate the psychometric properties of Sinhala translated version of ESRD-AQ for use among patients receiving haemodialysis in a Sri Lankan hospital.

## Material and methods

### Study design and setting

The study was a validation study which consisted of translation, cultural adaptation and evaluation of psychometric properties (face, content and construct validity) of the instrument. A descriptive cross-sectional study was conducted among Sinhala speaking patients receiving haemodialysis in Teaching Hospital, Kurunegala, Sri Lanka to evaluate construct validity of the instrument. Teaching Hospital, Kurunegala which is located nearly 100 km away from the capital city of Sri Lanka, has one of the main government sector haemodialysis units. It provides haemodialysis treatment to approximately forty to fifty patients daily during morning, afternoon and evening shifts. In this haemodialysis unit, there are patients coming from different backgrounds, different areas of the country and being treated for different etiologies (CKD of unknown origin, Diabetic nephropathy).

### The procedure of translation and cultural adaptation of original English version of ESRD-AQ into Sinhala language

Permission was obtained from the original authors who first designed and validated ESRD-AQ in California, United States prior to translation and validation in to Sinhala language. The translation of the instrument was performed according to the recommended guidelines provided by Beaton et al. on cross-cultural adaptation of self-reported measures [13]. The translation process included five steps- forward translation, synthesis of common version, back translation, expert committee consolidation and pretesting. The initial translation of ESRD-AQ from the original English language to the Sinhala language was done by two independent professional bilingual translators (language expert and subject expert). Both translations were compared by addition of another bilingual translator who was not involved in the forward translation process and discrepancies were resolved prior to synthesizing common Sinhalese version of the instrument. The forward translated common Sinhalese version of the instrument was independently back translated to the original English language by two other independent professional bilingual translators (language expert and subject expert). Both translations were compared by addition of another bilingual translator who was not involved in the back-translation process and discrepancies were resolved prior to synthesizing common English version. A committee consisting of experts in questionnaire designing and subject experts in the field of nephrology, reviewed both Sinhalese and English translations with original English version for semantic, idiomatic, and conceptual equivalence. The committee discussed and resolved the discrepancies of the instruments and reached the consensus on all forty-six items to retain in the instrument. A pre final version of the SINESRD-AQ was produced. Proof reading of the questionnaire was done by a multidisciplinary panel of experts in the field of Community Medicine, Nephrology, Nursing and Language.

### Evaluation of psychometric properties of SINESRD-AQ

Content validity of SINESRD-AQ was evaluated by obtaining opinion of experts (n = 6) in the research field and having extensive clinical background of treating and caring the patients with ESRD and haemodialysis (Nephrologist, Clinician, University lecturers, Nutritionist and Haemodialysis nurse) using modified Delphi technique. All the experts had more than five years of

experience in the relevant field and/or PhD or equivalent academic qualifications. The modified Delphi technique consisted of two rounds; Emailing of survey administration of the questionnaire to obtain feedback and final face-to-face meeting with experts to obtain consensus to produce final content validated questionnaire. Minor alterations were made in the sets of answers of SINESRD-AQ as suggested by the expert panel to make it compatible with the Sri Lankan setting. To ensure that the questionnaire had appropriate items or domains to represent the construct of interest, the expert panel assessed content relevance of each question, appropriateness of the words, their cultural relevance and domain coverage of adherence by each question on a 4-point Likert- type scale (1 = not relevant; 2 = somewhat relevant; 3 = quite relevant; 4 = highly relevant). Based on expert opinion, Item-level Content Validity Index (I-CVI), Average Item-level Content Validity Index for the whole scale (S-CVI) [14], Item-level Content Validity Ratio (I-CVR) [15] and Modified kappa statistic coefficient (k) [16] for each item were calculated. I-CVI $\geq$ 0.79 [17, 18], I-CVR$\geq$ 0.49 and k$\geq$ 0.74 [17, 18] were considered in retaining an item as 'valid'.

The resulting version was administered to a purposively selected group of patients (n = 10) receiving haemodialysis to evaluate face validity. The patients were asked to comment on each item of the instrument to determine any difficulty in understanding the words of the items or answer the items. In fact, their feedback was used to improve comprehensibility of the items of the instrument. Based on comments of the participants, no major modifications were indicated. However, based on the responses of the participants, a minor modification of the responses were made in the item 12, 23, 33 and 42 by combining the two responses 'Because I had an experience that I was sick after I missed the particular treatment (dialysis, medications, fluid restrictions or prescribed diet) / I was hospitalized after I missed the particular treatment'. These responses were combined because all the patients who missed the treatment due to a sickness admitted to the hospital following their sickness. The final SINESRD-AQ was produced, and proof reading was carried out to eliminate the spellings, grammatical and formatting errors prior to administering the questionnaire to the patients in order to evaluate construct validity.

## Sample and sample size calculation

The descriptive cross-sectional study included all the patients above the age of 18 years, diagnosed with ESRD, receiving haemodialysis at least for six consecutive months and able to speak Sinhala language from all ethnic groups attending haemodialysis at Teaching hospital, Kurunegala, Sri Lanka. Patients who were critically ill were excluded from the study. The rule of thumb, 5–10 subjects per each item was considered when calculating the sample size [19]. Considering 14 items in two subscales measuring adherence in ESRD-AQ, calculated sample size for the current study was 140 (1:10). Nonetheless, 150 patients were recruited for the study as the final sample with considering nonresponse rate of 5%. Participants who fulfilled selection criteria were recruited until the sample size was achieved in order to evaluate construct validity.

## Measurements

ESRD-AQ is an instrument consisting of 46 items designed to measure self-reported treatment adherence behaviour in four dimensions: haemodialysis (HD) attendance (14 items), medication use (9 items), fluid restrictions (10 items) and diet (8 items). The remaining 5 items assess patient's clinical history. Responses of the ESRD-AQ utilize a combination of Likert type, multiple choice and binary responses (yes/no). ESRD-AQ is comprised of two subscales measuring direct adherence behaviour (6 items- 14,17,18,26,31 and 46) and patients' attitudes and

perceptions about treatment (8 items– 11,12,22,23,32,33,41 and 42) [8]. Scoring the items of direct adherence behaviour subscale was done using the scoring system introduced in the original study and higher scores denoted to better adherence (maximum score-1200). Scoring of attitude/perception subscale was done in numerical way ranging from 1 (very good) to 5 (poor). Therefore, in the attitude/perception subscale, higher scores denoted to poor attitudes (maximum score-20) or perception (maximum score-20). Moreover, ESRD-AQ is a clinically validated instrument and widely used in different countries to assess treatment adherence behaviours of the patients receiving haemodialysis. Based on the responses to the items of direct adherence behaviour subscale of ESRD-AQ (Kim et al., 2010), adherence to each treatment modality of haemodialysis was categorized into good, moderate and poor as follows; If patient had not missed more than one haemodialysis session during the last month was considered as good adherence to haemodialysis attendance while if patient missed two haemodialysis sessions as moderate adherence and more than two sessions was considered as poor adherence without any medical reason. If patient never or very rarely missed prescribed medications during the last week was considered as good adherence to medications, if patient had missed medications half of the times as moderate adherence and most of the times or every time was considered as poor adherence. With regard to fluid and diet adherence, if patient followed the medical advice every time or most of the times considered as good adherence, half of the times as moderate adherence and very rarely or never as poor adherence.

Attitudes of the participants on each treatment modality was further assessed and considered as good if they answered as 'I fully understand that my kidney condition requires it' or 'because it is important to keep my body healthy'. If patient answered as 'because a medical professional told me to do so, because I was sick or hospitalized when I missed the treatment or I don't think that the treatment is very important to me' categorized as poor attitude. With regard to perception, good perception was classified as if patients mentioned that the particular treatment as extremely or highly important, moderate perception as if patient stated that the treatment was moderately important and as poor perception if patient mentioned the treatment has a least or no importance [6].

## Data collection procedure

Data collection was carried out from November 2019 to April 2020. Patients who were willing to participate in the study were recruited after obtaining written informed consent. Subsequently, data were collected using interviewer administered SINESRD-AQ and using a self-designed questionnaire for socio-demographic characteristics regularly by the principal investigator without disturbing the haemodialysis treatment they received. An independent sample of ten patients receiving haemodialysis completed the questionnaire twice with a retest interval of one week in order to assess test-retest reliability of the instrument.

## Data analysis

Descriptive statistics were analyzed using IBM Statistical Package for Social Sciences (SPSS) version 25.0 software for windows. The indices evaluating content validity of the instrument were calculated in Microsoft office excel using relevant formula. Reliability of two subscales were assessed with test-retest reliability with intra class correlation coefficient (ICC)>0.75 [20].

**Confirmatory factor analysis.** Construct validity of the SINESRD-AQ was assessed by confirmatory factor analysis (CFA) since the ESRD-AQ has strong theoretical evidence regarding its effective use to assess treatment adherence of the patients receiving haemodialysis in other countries [8–12]. Since all the items in two subscales were clinically validated in assessing

treatment adherence behaviours of the patients receiving haemodialysis, authors did not attempt to perform exploratory factor analysis in order to change or remove the items denoted to each subscale. Therefore, authors hypothesized two subscales (6- item direct adherence behaviour subscale and 8-item attitude/perception subscale) as found in the original English version and other language versions in performing CFA to verify the subscale validity and the existing measurement theory in assessing treatment adherence behaviours in the ESRD-AQ. CFA was further performed to identify how each item theoretically related to their respective subscale (latent construct/factor).

Prior to performing factor analysis, the data set was assessed for suitability, quality, missing data, and for violations of the assumptions of analytical techniques of CFA. No missing data or extreme outliers were observed. The normality of data was assessed with histograms and Q-Q plots, showed non-normal distribution. Both Shapiro-Wilk test and Kolmogorov–Smirnov test were significant at p value, 0.000 (p<0.05) for all the variables in the data set which rejects the null hypothesis for the normality for the sample at a 5% level of significance assuming non-normality of data. Four items with skewness value exceeding ±2 and four items with kurtosis value exceeding ±7 [21] were observed indicating non-normal distribution of data. The sampling adequacy was assessed using the Kaiser-Meyer-Olkin (KMO) test. A value 0.722 indicating that the sample size is adequate to perform factor analysis with 14-items of SINESR-D-AQ. Bartlett's test of sphericity was less than 0.001 indicating an adequate amount of collinearity between the items [22]. CFA, using polychoric correlations and Robust Unweighted Least Squares (RULS) estimation method in LISREL 11.0 was employed to estimate model parameters. RULS is especially designed for ordinal data and adjusted to non-normality. In fact, it yields better results in lower sample sizes and estimates parameters with less standard error compared to the other estimation methods [23]. The following goodness of fit indices were used: Normed Chi-square ($x^2$/df), Root Mean Squared Error of Approximation (RMSEA), Standardized Root Mean Square Residual (SRMR), Goodness of Fit Index (GFI), Adjusted Goodness of Fit Index (AGFI), Comparative Fit Index (CFI) and Non-Normed Fit Index (NNFI). $x^2$/df <5 [24, 25], RMSEA<0.1 [25], SRMR<0.08 [26] and GFI, CFI, NNFI >0.95 [25, 26] were considered as reference values for acceptable model fit.

## Ethical considerations

The Ethical approval was granted from Ethics Review Committee, Faculty of Medical Sciences, University of Sri Jayewardenepura, Sri Lanka (ERC No.06/19) and Committee of Ethics Review on Scientific Research, Teaching hospital, Kurunegala, Sri Lanka (THK/CERSR/2019/12) before commencement of the study. All the procedures of the study were performed in accordance with the recommended guidelines and regulations of the Declaration of Helsinki. Written and verbal informed consent was obtained from all the patients prior to commencement of data collection of the study.

## Results

### Socio-demographic characteristics and descriptive statistics of direct adherence behaviour subscale and attitude/perception subscale of SINESRD-AQ

The mean age of the study participants was 54.08±10.78 (±SD years) (Age range 18–74 years). Of the 150 participants, majority were males (72.7%), married (96.0%), Sinhalese (98.0%), educated up to ordinary level (grade six to eleven in school) (52.7%) and were not employed (86.0%). More than 70% (71.3%) of the patients were diagnosed to have chronic kidney disease

for less than five years and the majority (64.7%) were receiving haemodialysis for less than one year.

The mean overall adherence score in the present study was 990.17 (SD = ±106.10) ranging between 600–1200 (Table 1). Of the four treatment modalities, good adherence to HD attendance (did not miss or missed 1 dialysis session) was the highest (96.7%) among participants while good adherence to fluid restriction recommendations was the lowest (32.0%). Percentage of good adherence reported with prescribed medication and diet among participants were 96.0% and 71.3% respectively. Good perception towards following prescribed medications was the highest (98.7%) among participants and less than 15% had good attitude on all treatment modalities (Table 2).

## Psychometric properties of SINESRD-AQ

**Reliability analysis.** Strong test-retest reliability existed across all six items measuring adherence behaviour (direct adherence behaviour subscale) in SINESRD-AQ with intra class correlation coefficient of 0.837 (95% CI; lower bound-0.478; upper bound-0.957; p<0.001). Intra class correlation coefficient of attitude/perception subscale was 0.752 (95% CI; lower bound-0.274; upper bound-0.932; p<0.001).

**Face and content validity.** I-CVIs for the forty-six items of the final SINESRD-AQ were ranged between 0.83–1.00. Average I-CVI for the whole SINESRD-AQ (S-CVI) was 0.93. I-CVR were ranged between 0.67–1.00. Modified kappa statistic coefficients were ranged between 0.81–1.0, demonstrated good content validity of the questionnaire (Table 3).

**Table 1. Mean adherence score for various modalities of haemodialysis treatment regimen obtained from direct adherence behaviour subscale.**

| Item no. | Treatment modality | *Scoring for each response (As was given in original English version of ESRD-AQ) | Range of Score in present study | Mean adherence score ± SD |
|---|---|---|---|---|
| 14 | **HD attendance** (During last month, how many dialysis treatments did you miss completely?) | None (300), Missed one HD session (200), Missed two HD sessions (100) | 0–300 | 289.33± 46.19 |
| | | Missed three HD sessions (50) | | |
| | | Missed more than four HD sessions (0) | | |
| 17 | **Shortening of HD session** (During last month, how many times you have you shortened your dialysis time?) | Not applicable (200), Once (150) | 50–200 | 194.33±24.26 |
| | | Twice (100), Three times (50) | | |
| | | More than three times (0) | | |
| 18 | **Duration of shortening the HD session** (what was the average number of minutes?) | Not applicable (100) | 0–100 | 95.83±17.24 |
| | | <10 min or 10 min (75), 11 to 20 min (50), 21 to 30 min (25), >30 min (0) | | |
| 26 | **Adherence to medication** (during the past week, how often have you missed your prescribed medicines?) | None of the time (200), Very seldom (150), About half of the time (100) | 50–200 | 184.00 ±30.83 |
| | | Most of the time (50), All of the time (0) | | |
| 31 | **Adherence to fluid restrictions** (during the past week, how often have you followed fluid restriction recommendations?) | All of the time (200) | 0–200 | 92.67 ±59.75 |
| | | Most of the time (150) | | |
| | | About half of the time (100) | | |
| | | Very seldom (50) | | |
| | | None of the time (0) | | |
| 46 | **Adherence to recommended diet** (during the past week, how often have you followed dietary recommendations?) | All of the time (200) | 0–200 | 134.00±41.50 |
| | | Most of the time (150) | | |
| | | About half of the time (100) | | |
| | | Very seldom (50) | | |
| | | None of the time (0) | | |
| | **Total adherence score** | | 600–1200 | 990.17±106.10 |

**Table 2. Descriptive statistics of attitude/perception subscale.**

| Item No. | Variable | Level of attitudes/perception | | |
|---|---|---|---|---|
| | | Good % | Moderate % | Poor % |
| 11 | How important do you think it is to follow your dialysis schedule? | 94.7 | 4.0 | 1.3 |
| 22 | How important do you think it is to take your medicines as scheduled? | 98.7 | 0.7 | 0.7 |
| 32 | How important do you think it is to limit your fluid intake? | 93.3 | 5.3 | 1.3 |
| 41 | How important do you think it is to watch your diet daily? | 92.7 | 6.0 | 1.3 |
| 12 | Why do you think it is important to follow your dialysis schedule? | 13.3 | 29.3 | 57.3 |
| 23 | Why do you think it is important to take your medicines as scheduled? | 14.0 | 43.3 | 42.7 |
| 33 | Why do you think it is important for you to limit your fluid intake? | 12.0 | 30.7 | 57.3 |
| 42 | Why do you think it is important for you to watch your diet daily? | 12.0 | 53.3 | 34.7 |

Perception–Item 11,22,32,41; attitudes– 12,23,33,42

**Construct validity.** *CFA of direct adherence behaviour subscale.* Initially two basic models (6-item one factor and 6-item two factor) were tested using CFA. In addition to the basic models, 5-item model was hypothesized removing item no.18 ('During the last month, when your dialysis treatment was shortened, what was the average number of minutes?') from the analysis since it is a supplementary question to further elaborate the question No.17 ('how many times have you shortened your dialysis time?'). Permission was allowed to correlate error terms in the models as suggested by the software in order to achieve model fit. Of the four models, 5-item 2 factor model was come up with fairly adequate goodness of fit indices (Table 4) with factor loadings greater than 0.5 except item 46 (0.26) (Table 5).

*CFA of attitude/perception subscale.* 8-item attitude/perception subscale was also tested for two models– 8-item one factor and 8-item two factor. Of the two models, two factor model was shown improved model fit (Table 6) with factor loadings of all the items greater than 0.5 (Table 7).

## Discussion

ESRD-AQ is one of the widely used instruments to determine self-reported adherence among patients receiving haemodialysis. It has been translated, culturally adapted and validated into different languages in different countries. The main objective of this study was to evaluate the psychometric properties of Sinhala translated version of ESRD-AQ. The findings of the present study have supported the structure of the measurement theory and the content validity of the SINESRD-AQ in measuring treatment adherence behaviours of the patients receiving haemodialysis in a Sri Lankan hospital.

Present study found that the content validity of SINESRD-AQ was excellent with I-CVI for all the forty-six items between 0.83–1.0 ($\geq$0.79), S-CVI of 0.93, I-CVR between 0.67–1.0 ($\geq$0.49) and Modified kappa statistic >0.8 ($\geq$0.74). These findings are consistent with the findings of previous studies such as original English version (I-CVI = 0.86–1.00; S-CVI-0.99) [8], Spanish version (I-CVI = 0.97–0.99; S-CVI = 0.98) [10], Portuguese version (I-CVI = 0.94–1.00;S-CVI = 0.98) [9] and Brazilian version (I-CVI>0.8; S-CVI = 0.96) [11]. "I-CVI does not consider the inflated values that may occur because of possibility of chance agreement" [16]. Therefore, kappa statistic was calculated for all the forty-six items as it is "a consensus index of interrater agreement that supplements CVI". In fact, kappa index ensures that the agreement among experts is beyond chance. "Kappa values above 0.74 are considered as excellent, between 0.60 and 0.74 as good, and between 0.40 and 0.59 are considered as fair"

**Table 3. Content validation of 46-items of SINESRD-AQ.**

| Item | No.of experts in agreement/ Total number of experts (n = 6) | I-CVI | Interpretation (≥0.79) | I-CVR | Interpretation (≥0.49) | Modified Kappa statistic | Interpretation (≥0.74) |
|------|---|---|---|---|---|---|---|
| 1. | 6 | 1 | Appropriate | 1 | Remained | 1 | Excellent |
| 2. | 5 | 0.83 | Appropriate | 0.67 | Remained | 0.816092 | Excellent |
| 3. | 5 | 0.83 | Appropriate | 0.67 | Remained | 0.816092 | Excellent |
| 4. | 5 | 0.83 | Appropriate | 0.67 | Remained | 0.816092 | Excellent |
| 5. | 5 | 0.83 | Appropriate | 0.67 | Remained | 0.816092 | Excellent |
| 6. | 6 | 1 | Appropriate | 1 | Remained | 1 | Excellent |
| 7. | 5 | 0.83 | Appropriate | 0.67 | Remained | 0.816092 | Excellent |
| 8. | 5 | 0.83 | Appropriate | 0.67 | Remained | 0.816092 | Excellent |
| 9. | 6 | 1 | Appropriate | 1 | Remained | 1 | Excellent |
| 10. | 6 | 1 | Appropriate | 1 | Remained | 1 | Excellent |
| 11. | 6 | 1 | Appropriate | 1 | Remained | 1 | Excellent |
| 12. | 6 | 1 | Appropriate | 1 | Remained | 1 | Excellent |
| 13. | 6 | 1 | Appropriate | 1 | Remained | 1 | Excellent |
| 14. | 6 | 1 | Appropriate | 1 | Remained | 1 | Excellent |
| 15. | 6 | 1 | Appropriate | 1 | Remained | 1 | Excellent |
| 16. | 6 | 1 | Appropriate | 1 | Remained | 1 | Excellent |
| 17. | 5 | 0.83 | Appropriate | 0.67 | Remained | 0.816092 | Excellent |
| 18. | 5 | 0.83 | Appropriate | 0.67 | Remained | 0.816092 | Excellent |
| 19. | 5 | 0.83 | Appropriate | 0.67 | Remained | 0.816092 | Excellent |
| 20. | 6 | 1 | Appropriate | 1 | Remained | 1 | Excellent |
| 21. | 6 | 1 | Appropriate | 1 | Remained | 1 | Excellent |
| 22. | 6 | 1 | Appropriate | 1 | Remained | 1 | Excellent |
| 23. | 6 | 1 | Appropriate | 1 | Remained | 1 | Excellent |
| 24. | 6 | 1 | Appropriate | 1 | Remained | 1 | Excellent |
| 25. | 5 | 0.83 | Appropriate | 0.67 | Remained | 0.816092 | Excellent |
| 26. | 6 | 1 | Appropriate | 1 | Remained | 1 | Excellent |
| 27. | 6 | 1 | Appropriate | 1 | Remained | 1 | Excellent |
| 28. | 5 | 0.83 | Appropriate | 0.67 | Remained | 0.816092 | Excellent |
| 29. | 5 | 0.83 | Appropriate | 0.67 | Remained | 0.816092 | Excellent |
| 30. | 5 | 0.83 | Appropriate | 0.67 | Remained | 0.816092 | Excellent |
| 31. | 6 | 1 | Appropriate | 1 | Remained | 1 | Excellent |
| 32. | 6 | 1 | Appropriate | 1 | Remained | 1 | Excellent |
| 33. | 6 | 1 | Appropriate | 1 | Remained | 1 | Excellent |
| 34. | 5 | 0.83 | Appropriate | 0.67 | Remained | 0.816092 | Excellent |
| 35. | 5 | 0.83 | Appropriate | 1 | Remained | 1 | Excellent |
| 36. | 5 | 0.83 | Appropriate | 0.67 | Remained | 0.816092 | Excellent |
| 37. | 5 | 0.83 | Appropriate | 0.67 | Remained | 0.816092 | Excellent |
| 38. | 5 | 0.83 | Appropriate | 0.67 | Remained | 0.816092 | Excellent |
| 39. | 6 | 1 | Appropriate | 1 | Remained | 1 | Excellent |
| 40. | 6 | 1 | Appropriate | 1 | Remained | 1 | Excellent |
| 41. | 6 | 1 | Appropriate | 1 | Remained | 1 | Excellent |
| 42. | 6 | 1 | Appropriate | 1 | Remained | 1 | Excellent |
| 43. | 5 | 0.83 | Appropriate | 0.67 | Remained | 0.816092 | Excellent |
| 44. | 6 | 1 | Appropriate | 1 | Remained | 1 | Excellent |
| 45. | 6 | 1 | Appropriate | 1 | Remained | 1 | Excellent |
| 46. | 6 | 1 | Appropriate | 1 | Remained | 1 | Excellent |

**Table 4. Goodness of fit indices for various models of direct adherence behaviour subscale.**

| Model | Fit indices | | | | | | | |
|---|---|---|---|---|---|---|---|---|
| | Chi-squared-$x2$ (df) | $x^2$/df | RMSEA | SRMR | CFI | NNFI | GFI | AGFI |
| 6-item 1-factor | 82.70 (8) | 10.33 | 0.250 | 0.0620 | 0.869 | 0.755 | 0.985 | 0.962 |
| 6-item 2-factor | 72.47 (6) | 12.07 | 0.273 | 0.0407 | 0.884 | 0.709 | 0.994 | 0.978 |
| 5-item 1-factor | 31.51 (4) | 7.88 | 0.215 | 0.0668 | 0.858 | 0.645 | 0.980 | 0.927 |
| **5-item 2-factor** | 17.21 (3) | **5.73** | **0.178** | **0.0415** | **0.927** | **0.756** | **0.992** | **0.962** |

**Table 5. Factor loadings (standardized) for various models of direct adherence behaviour subscale.**

| Item No. | 6-item model | | 5-item model | |
|---|---|---|---|---|
| | **1-factor** | **2-factor** | **1-factor** | **2-factor** |
| 14 | 0.80 | 0.80 (F1) | 0.71 | 0.99(F1) |
| 17 | -0.95 | -0.95 (F1) | -1.10 | -0.78(F1) |
| 18 | -0.98 | -0.98(F1) | Removed | |
| 26 | 0.23 | 0.75 (F2) | 0.20 | 0.98(F2) |
| 31 | 0.14 | 0.49 (F2) | 0.55 | 1.34(F2) |
| 46 | 0.10 | 0.28 (F2) | 0.65 | 0.26(F2) |

2-factor model–Factor 1(F1) -Adherence to haemodialysis treatment; Factor 2(F2)—Adherence to other treatment modalities (medication, fluid restriction and diet)

**Table 6. Goodness of fit indices for various models of attitude/perception subscale.**

| Model | Fit indices | | | | | | | |
|---|---|---|---|---|---|---|---|---|
| | Chi-squared-$x2$ (df) | $x^2$/df | RMSEA | SRMR | CFI | NNFI | GFI | AGFI |
| 8-item 1-factor | 176.137 (20) | 8.80 | 0.229 | 0.094 | 0.758 | 0.661 | 0.970 | 0.945 |
| **8-item 2-factor** | **89.674 (19)** | **4.72** | **0.158** | **0.0543** | **0.890** | **0.839** | **0.990** | **0.981** |

[16]. In the present study, Kappa statistic coefficient higher than 0.8, demonstrated excellent content validity of SINESRD-AQ. I-CVR measures essentiality of each item in a scale -1 to +1 and higher scores are denoted to greater agreement among the experts. I-CVR greater than 0.49 is the acceptable level of significance in each item to be remained [18] and in the present

**Table 7. Factor loadings (standardized) of attitude/perception subscale of ESRD-AQ.**

| Item No. | 1-factor model | 2-factor model |
|---|---|---|
| 11 | 0.49 | 0.55 (F1) |
| 12 | 0.75 | 0.79 (F2) |
| 22 | 0.63 | 0.82 (F1) |
| 23 | 0.91 | 0.96 (F2) |
| 32 | 0.55 | 0.62 (F1) |
| 33 | 0.65 | 0.70 (F2) |
| 41 | 0.69 | 0.79 (F1) |
| 42 | 0.75 | 0.79 (F2) |

2-factor model–Factor 1(F1) -Perception of importance towards treatment regimen; Factor 2(F2)–attitudes about various treatment modalities

study, all forty-six items had I-CVR > 0.6 showing acceptable content validity of the SINESRD-AQ.

In the original English version, Cronbach's Alpha was omitted since the instrument design does not consist homogenous items to measure internal consistency reliability [8]. Thus, ESR-D-AQ favors alternative and more powerful reliability measurements due to its' limitation in calculating Cronbach's Alpha. In the original English version, test-retest reliability with ICC value which is one of the best assessments that measures the reproducibility of consistent test scores by the same person on different occasions over a short time interval [27] was used to confirm the reliability of the instrument. According to Koo and Li (2016), ICC value less than 0.5, between 0.5–0.75, 0.75–0.9 and greater than 0.9 are indicative of poor, moderate, good and excellent reliability respectively [20]. In this study, good test-retest reliability was confirmed across all the 6 items measuring direct treatment adherence behaviours with average ICC of 0.837 (95% CI from 0.478–0.957; p<0.01) and 8 items assessing attitude/perception of SINESRD-AQ with average ICC of 0.752 (95% CI from 0.274–0.932; p<0.01). Similarly, the strong test-retest reliability was observed in other language versions of ESRD-AQ with ICC of 0.91(from 0.83–1.00 with 95% CI) in original English version, 0.93 (from 0.694–0.984 with 95% CI) in Portuguese version and 0.96 (from 0.82–1.00 with 95% CI) in Spanish version. Besides, the excellent ICC of 0.98 and 0.91 was reported for adherence behaviour subscale and attitude/perception subscale respectively in Brazilian version.

Previous studies found in the literature have assessed construct validity of ESRD-AQ by employing known groups analysis by comparing the group of adheres with group of non-adheres. Adheres and non-adheres were distinguished based on the responses to the items directly measuring adherence in ESRD-AQ by the patients and respective clinical measures, i.e. "Patients were considered as non-adherent to their medications and diet if their serum phosphorous level is higher than 7.5mg/dl" [8]. Although the several studies have proven scientific validity and reliability of ESRD-AQ with different methods, none of the studies have examined the validity and reliability of ESRD-AQ by employing factor analysis.

"Confirmatory factor analysis is typically used in later phases of scale development or construct validation after the underlying structure has been tentatively established by prior empirical analysis using exploratory factor analysis as well as on theoretical grounds" [28]. The items of two subscales of SINESRD-AQ were not exploratory in nature and they directly measure different aspects of adherence of the patients (e.g. HD attendance, medication adherence, fluid adherence, diet adherence). Therefore, it was not required to explore underlying factor structure by removing the items from the scale. Since the originally proposed subscales are clinically validated and extensively used in measuring adherence, present study attempted to verify its applicability for the Sinhala speaking patients receiving haemodialysis conducting CFA using standard covariance based Structural Equation Modelling. It was interesting to note that of the items measuring direct treatment adherence, all the items showed positive effect on treatment adherence despite item no.17 and 18 which show negative effect with negative factor loadings. This means when the frequency and duration of shortening of the dialysis treatment is increased, the adherence to haemodialysis is better among participants. The reason for this could be that only a small number of participants mentioned that they shortened their HD session without a medical reason, resulting in a lesser impact on positive treatment adherence.

Authors sequentially hypothesized different factor models for two subscales and tested their fit to the data set of the present study. Four models were tested for direct adherence behaviour subscale including 6 item one factor, 6 item two factor, 5 item one factor and 5 item two factor model. Two models were evaluated for attitude/perception subscale including 8 item one factor and 8 item two factor model. In both subscales, two factor model (5 item and 8 item) was shown comparatively acceptable model fit. However, of the goodness of fit indices, RMSEA

and chi-square values were unsatisfactory in both factor models while other indices showed acceptable model fit. According to the Kenny et al. (2015), for models with small *df*, the RMSEA value can exceeds the cut off value very often leading to poor model fit [29]. Therefore, it was suggested not to compute RMSEA for very low *df* models that do not have large sample size. Besides, it often results in rejecting correctly specified or closely fitted models. In present study, both subscales had small *df* while direct adherence behaviour subscale had extremely low value (5-item two factor model-*df* = 3). Since SRMR and CFI are less susceptible to the effect of *df*, they can be used to interpret models with small *df* instead of RMSEA. In fact, compared to RMSEA, SRMR shows higher power to reject non-close fit models in lower sample sizes less than 200 [30]. Hu and Bentler (1999) recommended a two-index strategy using a combination of SRMR<0.09 with another supplementary index including either NNFI (TLI)>0.96, RMSEA<0.06 or CFI>0.96 when evaluating model fit [26]. In the present study in both direct adherence behaviour subscale (SRMR = 0.0415; CFI = 0.927) and attitude/perception subscale (SRMR = 0.0543; CFI = 0.890), showed closure values to the thresholds of SRMR and CFI indicating fairly acceptable model fit.

The current study has a few limitations. The sample size of the present study was small (n = 150) and the future studies should be conducted with a large sample size for further confirmation of the subscale validity and the factor structure of SINESRD-AQ. This validation study was carried out in a single haemodialysis center in Sri Lanka. Hence, a multicenter study is encouraged in the future studies to capture maximum variation of the study sample.

## Conclusion

The validated SINESRD-AQ can be utilized as a multidimensional instrument to assess treatment adherence behaviours of the Sinhala speaking patients receiving haemodialysis periodically. Therefore, any deficiencies identified with adherence could be intervened by the health care staff, thereby preventing the occurrence of adverse events and to improve better patient outcomes. In fact, it enables planning of care while making appropriate changes and adaptations to the prescribed treatment regimen. The present study concludes that the SINESRD-AQ is a scientifically and statistically valid and reliable instrument to assess treatment adherence behaviours among Sinhala speaking patients receiving haemodialysis in a Sri Lankan hospital.

## Supporting information

**S1 File. Sinhalese version of End Stage Renal Disease-Adherence Questionnaire.**
(PDF)

**S2 File. Cultural adaptation of Sinhalese version of End Stage Renal Disease-Adherence Questionnaire.**
(PDF)

## Acknowledgments

The authors greatly acknowledge and appreciate the contribution of all the study participants and the staff of the haemodialysis unit, Teaching Hospital, Kurunegala, Sri Lanka. Dr. Chinthana Galahitiyawa (Consultant Nephrologist, Sri Jayewardenepura General Hospital), Dr. Dinesha Sudusinghe (Consultant Nephrologist/Lecturer, Faculty of Medical Sciences, University of Sri Jayewardenepura), Ms. Mala Abeygunawardena (Nutritionist, Sri Jayewardenepura General Hospital) and all the dialysis nurses who participated in the modified Delphi technique to evaluate content validity of the SINESRD-AQ are greatly appreciated for their valuable input. We wish to acknowledge the authors of original English version of ESRD-AQ,

Youngmee Kim (Professor), Lorraine S.Evangelista (Professor), Linda R.Phillips (Professor), Carol Pavlish (Associate Professor) and Joel D.Kopple (Professor) for granting permission to use ESRD-AQ for the present study.

## Author Contributions

**Conceptualization:** Chalani Lasanthika, Kamani Wanigasuriya, Usha Hettiaratchi, Thamara Dilhani Amarasekara, Christine Sampatha Evangeline Goonewardena.

**Data curation:** Chalani Lasanthika, Kamani Wanigasuriya, Usha Hettiaratchi, Thamara Dilhani Amarasekara, Christine Sampatha Evangeline Goonewardena.

**Formal analysis:** Chalani Lasanthika.

**Funding acquisition:** Christine Sampatha Evangeline Goonewardena.

**Investigation:** Chalani Lasanthika.

**Methodology:** Chalani Lasanthika, Kamani Wanigasuriya, Usha Hettiaratchi, Thamara Dilhani Amarasekara, Christine Sampatha Evangeline Goonewardena.

**Supervision:** Kamani Wanigasuriya, Usha Hettiaratchi, Thamara Dilhani Amarasekara, Christine Sampatha Evangeline Goonewardena.

**Writing – original draft:** Chalani Lasanthika.

**Writing – review & editing:** Chalani Lasanthika, Kamani Wanigasuriya, Usha Hettiaratchi, Thamara Dilhani Amarasekara, Christine Sampatha Evangeline Goonewardena.

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
