## [Decision Letter · Decision Letter 0]

5 Jul 2023

PONE-D-22-20481Psychometric properties of End Stage Renal Disease-Adherence Questionnaire-Sinhalese version among patients receiving haemodialysisPLOS ONE

Dear Dr. Lasanthika,

Thank you for submitting your manuscript to PLOS ONE. After careful consideration, we feel that it has merit but does not fully meet PLOS ONE’s publication criteria as it currently stands. Therefore, we invite you to submit a revised version of the manuscript that addresses the points raised during the review process.

We look forward to receiving your revised manuscript.

Kind regards,

Surangi Jayakody, MBBS, MSc, MD

Academic Editor

PLOS ONE

Journal Requirements:

Reviewers' comments:

Reviewer's Responses to Questions

**Comments to the Author**

1. Is the manuscript technically sound, and do the data support the conclusions?

Reviewer #1: Yes

Reviewer #2: Yes

2. Has the statistical analysis been performed appropriately and rigorously? 

Reviewer #1: Yes

Reviewer #2: Yes

3. Have the authors made all data underlying the findings in their manuscript fully available?

Reviewer #1: Yes

Reviewer #2: No

4. Is the manuscript presented in an intelligible fashion and written in standard English?

Reviewer #1: Yes

Reviewer #2: Yes

5. Review Comments to the Author

Reviewer #1: It is my opinion that the authors of this manuscript have presented the details of an extensive process of adapting and validating a questionnaire with potential relevance in clinical practice as well as research. Overall the manuscript is of good quality, however it could be further improved prior to being published. It is hoped that the below mentioned comments and suggestions could be of use in enhancing the paper.

1. Adaptation of the questionnaire

Even though the process of translation and cultural adaptation have been well detailed in the Methodology section, nothing with regard to the qualitative components in these steps have been mentioned in the Results section. What this means is, the possible changes that had to be made to the questionnaire at each step before arriving at the final version, and why those were made. This is very important, especially, from a cultural adaptation standpoint. For example; it was noted that in SINESRD-AQ the answer options in questions 12. 23, 33 and 42 had been modified by combining the ‘got sick’ and ‘hospitalised’ options from the original ESRD-AQ. It would be important to specify, in the paper, such changes and reasons behind them. This modification has resulted in changes to the possible scores of the items and subscales. Similarly, changes in wording must have been required to make it suitable for the Sri Lankan setting.

2. Utilisation of the findings of the CFA

Based on the CFA findings, the authors have proposed that a 5-item 2-factor model is the best fit for Adherence Behaviour subscale, and an 8-item 2-factor is most appropriate for Attitude/Perception subscale. However, this has not resulted in the authors making (or at least suggesting) any changes to the final questionnaire or the way it should be scored. Since each item has its individual importance, removing them based on the CFA interpretation may not be appropriate in this type of questionnaire. But, when a scale score is calculated, the items that should be used and the scoring method may have to be modified in the face of such findings.

3. Presence of negative factor loadings in Adherence Behaviour subscale

All the items assessing adherence in SINESRD-AQ were coded to have higher values for higher adherence. Yet, it is interesting to note that factor loadings of all the CFA models are negative for items related to shortening HD sessions (items 17 and 18). If we assume that the latent factor is for positive treatment adherence, HD session shortening is acting in the opposite direction. This seems counter intuitive. Even though I am not experienced in practicalities of HD session in Sri Lanka, my suspicion is that patients have little or no say in determining the length of sessions. Considering the authority of health staff in the Sri Lankan healthcare setup, these questions could be irrelevant or incorrectly interpreted by the patients.

4. Rationale of using factor analysis in a questionnaire such as this

It seems like a bold decision by the authors to utilise factor analysis for this type of questionnaire, while it had not been used in any of the previous publications on ESRD-AQ given as references. As noted in the manuscript itself, the original paper by Kim et al. (2010) notes that the instrument would not be suitable to assessed using a measure of internal consistency such as Cronbach’s Alpha. Then again, it uses the summing of item scores to produce subscale scores, which suggests the presence of an underlying common construct that is being measured by the selected items. Factor analysis assumes the presence of latent variables (factors) represented by one or more of the scale items. Considering that each item of SINESRD-AQ measures a specific aspects of CKD management, the need of factor analysis in this study could be questioned.

5. Uniformity on usage of the second subscale name

The second subscale mentioned in the paper seem to take two forms: Attitude/Perception and Attitude/Knowledge. But, it must be stated that this error was seen in the original paper by Kim et al. (2010) as well. I would suggest sticking to one of the two, preferably Attitude/Perception, as knowledge does not seem to be what the questions measure.

6. Reclassification of item responses

In this, the focus is on lines from 255 to 260 (on pages 12 and 13) as well as Table 2 on page 14. It has not been explained how these percentages were arrived at. What does it mean to have 96.7% HD adherence when the relevant question (14) only has answers on the number of HD treatments that were missed? Similarly, for the other values noted in page 12 and Table 2. There seem to have been a reclassification of responses performed. But this has not been mentioned in the methodology or results section.

7. Errors in writing

Even though the overall writing of the manuscript is of good quality, there are a fair amount of errors that have to be corrected. It would be difficult to point out each. A proper and detailed proofreading would be required. Some of them are noted below:

• Lines 51 to 53 on page 3 – This sentence is not clear. Perhaps stating it as “… 2003 to 2013 had increased by over eight thousand per year with around thousand haemodialysis sessions per month in 2013”.

• Lines 84 to 86 in page 4 and 5 – This is an improperly structured sentence.

• Line 105 on page 5 – Shouldn’t it be “CKD of unknown origin”?

• Line 125 on page 6 – Shouldn’t it be “A committee consisting of…”?

• Line 249 on page 12 - up to ordinary level could mean any level of education up to that level. Shouldn’t this be Grade 5 up to Ordinary Level?

8. Errors in references

There were few mistakes in referencing of the manuscript. It would be good to recheck them even if a referencing software was used. Some of the noted errors are:

• Line 75 on page 4 – Even though Palestine is mentioned, the references does not seem to give results from a Palestinian study.

• Line 149 on page 7 – the reference 17 appears twice. It seems that the second one should be 18, as the next reference noted in the text is 19.

• Line 202 on page 10 – Reference 3 does not seem to be relevant here. Order of references should be 8, 9, 10, …

• Lines 534 and 535 on page 28 – The reference is incomplete

• Line 541 on page 28 – Mixing up of page number and DOI

• Lines 570 to 573 on pages 29 and 30 – Some components in full capital

• Line 575 on page 30 - incomplete reference

Reviewer #2: The team of authors have done a commendable work in translating and analysing the psychometric properties of ESRD adherence questionnaire - Sinhalese version. Addressing the comments below will further improve the scientific rigor of the paper.

1 Line 49

//Nonetheless, haemodialysis has become as the most efficient and practical treatment modality among patients with ESRD in Sri Lanka due to resource constraints and shortage of kidney donors for transplantation [4]// Referred article does not discuss haemodialysis being the most efficient treatment for ESRD. Please correct.

2 Line 51

//It was observed that the number of dialysis sessions carried out in National Hospital of Sri Lanka from 2003 to 2013 were increased approximately over eight thousand with thousand haemodialysis sessions per month// Does this refer to haemodialysis or all types of dialyses together?

3 Line 152

//The resulting version was administered to a purposively selected group of patients (n=10)

receiving haemodialysis to evaluate face validity// Please explain which characteristics of the participants were considered in ‘purposively selection’ and why those characteristics were considered.

4 Line 187

//data were collected using interviewer administered SINESRD-AQ// Please explain why the study instrument was administered through an interviewer, when it’s designed to be used as self-reported tool.

5 Line 379

How do you justify using Cronbach’s Alpha to assess internal consistency of the translated study instrument when the original instrument has omitted it due to instrument design does not consist homogenous items to measure internal consistency reliability?

6 Number of tables - Please check with journal guidelines for the number of tables/ figures allowed.

6. PLOS authors have the option to publish the peer review history of their article (what does this mean?). If published, this will include your full peer review and any attached files.

Reviewer #1: No

Reviewer #2: No

---

## [Author Response · Author response to Decision Letter 0]

30 Aug 2023

We greatly appreciate the reviewers’ interest about the manuscript titled 'Psychometric properties of End Stage Renal Disease-Adherence Questionnaire-Sinhalese version among patients receiving haemodialysis’' and quality comments and suggestions offered by the reviewers for its further improvement. The observations of the reviewers gave a greater value to the manuscript and enhanced its validity and readability. The authors’ responses for the comments are given in the separate file submitted herewith. Thank you very much.

---

## [Decision Letter · Decision Letter 1]

3 Oct 2023

Psychometric properties of End Stage Renal Disease-Adherence Questionnaire-Sinhalese version among patients receiving haemodialysis

PONE-D-22-20481R1

Dear Chalani,

We’re pleased to inform you that your manuscript has been judged scientifically suitable for publication and will be formally accepted for publication once it meets all outstanding technical requirements.

Kind regards,

Surangi Jayakody, MBBS, MSc, MD

Academic Editor

PLOS ONE

Reviewers' comments:

Reviewer's Responses to Questions

**Comments to the Author**

1. If the authors have adequately addressed your comments raised in a previous round of review and you feel that this manuscript is now acceptable for publication, you may indicate that here to bypass the “Comments to the Author” section, enter your conflict of interest statement in the “Confidential to Editor” section, and submit your "Accept" recommendation.

Reviewer #1: All comments have been addressed

Reviewer #2: All comments have been addressed

2. Is the manuscript technically sound, and do the data support the conclusions?

Reviewer #1: Yes

Reviewer #2: Yes

3. Has the statistical analysis been performed appropriately and rigorously? 

Reviewer #1: Yes

Reviewer #2: Yes

4. Have the authors made all data underlying the findings in their manuscript fully available?

Reviewer #1: Yes

Reviewer #2: No

5. Is the manuscript presented in an intelligible fashion and written in standard English?

Reviewer #1: Yes

Reviewer #2: Yes

6. Review Comments to the Author

Reviewer #1: Even though the specific writing errors mentioned as examples in review comments had been addressed, my suggestion to do a detailed proofreading of the manuscript to correct other errors appears not to have been attempted. I strongly recommend that the authors do a proper proofreading prior to the final submission. For the convenience of the authors, I have attached the manuscript with additional writing errors HIGHLIGHTED IN BLUE. It is hoped that the authors would be able to figure out the reason for marking each point as an error, and correct if suitable.

Reviewer #2: The authors have adequately responded to all the queries raised by the reviewers and I'm happy to accept the revised paper for publication. Congratulations.

7. PLOS authors have the option to publish the peer review history of their article (what does this mean?). If published, this will include your full peer review and any attached files.

Reviewer #1: No

Reviewer #2: No

---

## [Editor Report · Acceptance letter]

11 Oct 2023

PONE-D-22-20481R1 

Psychometric properties of End Stage Renal Disease-Adherence Questionnaire-Sinhalese version among patients receiving haemodialysis 

Dear Dr. Lasanthika:

I'm pleased to inform you that your manuscript has been deemed suitable for publication in PLOS ONE. Congratulations! Your manuscript is now with our production department. 

Kind regards, 

on behalf of

Dr Surangi Jayakody 

Academic Editor

PLOS ONE